# PeerJ

# Absence of *Helicobacter pylori* is not protective against peptic ulcer bleeding in elderly on offending agents: lessons from an exceptionally low prevalence population

Yeong Yeh Lee[1,2], Nordin Noridah[1], Syed Abdul Aziz Syed Hassan[1] and Jayaram Menon[3]

[1] School of Medical Sciences, Universiti Sains Malaysia, Kubang Kerian, Kelantan, Malaysia
[2] Section of Gastroenterology & Hepatology, Department of Medicine, Medical College of Georgia, Georgia Regents University, Augusta, Georgia
[3] Department of Medicine, Queen Elizabeth Hospital, Kota Kinabalu, Sabah, Malaysia

## ABSTRACT

**Aim.** *Helicobacter pylori* (*H. pylori*) infection is exceptionally rare in population from the north-eastern region of Peninsular Malaysia. This provides us an opportunity to contemplate the future without *H. pylori* in acute non-variceal upper gastrointestinal (GI) bleeding.

**Methods.** All cases in the GI registry with GI bleeding between 2003 and 2006 were reviewed. Cases with confirmed non-variceal aetiology were analysed. Rockall score $> 5$ was considered high risk for bleeding and primary outcomes studied were in-hospital mortality, recurrent bleeding and need for surgery.

**Results.** The incidence of non-variceal upper GI bleeding was 2.2/100,000 person-years. Peptic ulcer bleeding was the most common aetiology (1.8/100,000 person-years). In-hospital mortality (3.6%), recurrent bleeding (9.6%) and need for surgery (4.0%) were uncommon in this population with a largely low risk score (85.2% with score $\leq 5$). Elderly were at greater risk for bleeding (mean 68.5 years, $P = 0.01$) especially in the presence of duodenal ulcers ($P = 0.04$) despite gastric ulcers being more common. NSAIDs, aspirin and co-morbidities were the main risk factors.

**Conclusions.** The absence of *H. pylori* infection may not reduce the risk of peptic ulcer bleeding in the presence of risk factors especially offending drugs in the elderly.

Corresponding author
Yeong Yeh Lee, justnleeyy@gmail.com

## INTRODUCTION

Non-variceal upper gastrointestinal (GI) bleeding remains a prevalent condition and its mortality hardly change despite a declining trend of peptic ulcer disease and improvement in therapeutic approaches. The reported incidence from North America and Europe was 20–60/100,000 populations but data from Asia were unfortunately scarce and variable (*Sung et al., 2011*). A recent study from Thailand reported an incidence of

152.9/100,000 population (*Sangchan et al., 2012*) and data from East Malaysia (State of Sabah), available only in abstract, reported an incidence of 72/100,000 population (*Cheng et al., 2001*). Reports from two tertiary hospitals in central Peninsular Malaysia indicated an overall low prevalence of non-variceal upper GI bleeding among the ethnic Malays (*Lim et al., 2003*; *Lakhwani et al., 2000*; *Lee, Mahendra Raj & Graham, 2013*).

There is a reducing trend of peptic ulcer disease observed within Asia, and this is largely a result of reducing prevalence of *H. pylori* infection. This trend is likely to continue into the future and a time will come when *H. pylori* joins the ranks of smallpox and polio (*Graham, Yamaoka & Malaty, 2007*). The population in the north-eastern region of Peninsular Malaysia (state of Kelantan), that consists of 90% ethnic Malays, had a seroprevalence of *Helicobacter pylori* (*H. pylori*) infection of only 4.2% among 496 blood donors and 4.8% among 921 patients attending a health screening clinic (*Uyub et al., 1994*). The *H. pylori* infection rate reported from gastric biopsies was 20% in duodenal ulcer, 21.2% in gastric ulcer, 16.7% in duodenal erosion and 17.1% in gastric erosion (*Raj et al., 2001*). The incidence of peptic ulcer perforations within the region from 1991 to 1992 was only 1.5/100,000 person-years (*Uyub et al., 1994*).

The exceptional low prevalence of *H. pylori* in the population from north-eastern region of Peninsular Malaysia provides us with an opportunity to contemplate the future without the infection. Our study aimed to determine the risk and clinical outcomes of acute non-variceal upper GI bleeding in this population with a low prevalence of *H. pylori* infection. The association between clinical characteristics, risk factors and treatment given with risk and bleeding outcomes was also assessed.

## METHODS

### Study population

We reviewed and analysed all prospective cases with a diagnosis of GI bleeding between 2003 and 2006 in our GI registry database. Cases were admitted into a tertiary university hospital (Hospital Universiti Sains Malaysia) situated in the north-eastern region of Peninsular Malaysia (State of Kelantan). The region consisted of 0.7 to 0.8 million of population (2003–2006) with a diverse racial background but has a predominant Malay population of approximately 90%.

All adults above 18 years old with upper GI bleeding as a diagnosis in the GI registry were then screened for inclusion. Subjects with typical symptoms and signs and subsequently requiring upper endoscopy after informed consent and confirmed to have non-variceal causes of acute upper GI bleeding were then included into the analysis. Upper endoscopy was performed in all cases within 24 h upon admission. *H. pylori* status, where available, as detected by either a rapid urease test (CLOtest®, Kimberly-Clark, Roswell, USA) and or histology from biopsies taken during endoscopy, would also be recorded. Exclusion criteria included those patients with lower GI bleeding, variceal bleeding, bleeding due to underlying hematologic disorders, GI bleeding of unknown origin, and those patients who did not have an endoscopy examination.

The study was approved by the Human Ethics Committee of Universiti Sains Malaysia.

### Study outcome and definitions

Rockall score (*Rockall et al., 1996*) was utilised to classify study population into low risk (score ≤ 5) and high risk (score > 5) (*Bessa et al., 2006*) for non-variceal upper GI bleeding. Briefly, Rockall score is made up of five variables, three of which are clinical parameters (age, shock and co-morbidities) and the other two endoscopic features (causative lesions and stigmata of recent haemorrhage) (*Rockall et al., 1996*; *Forrest, Finlayson & Shearman, 1974*). Each variable can be scored between 0 and 3, with a maximum score of 15 for all 5 variables.

The primary study outcome was to determine risk, based upon the Rockall score, of in-hospital mortality, recurrent bleeding and the need for surgery in this population with non-variceal upper GI bleeding. Clinical characteristics, risk factors, endoscopic features and endoscopic treatment given were also assessed for their association with study outcomes.

In-hospital mortality was defined as death during the period of hospital stay which was directly associated with upper GI bleeding and this was compared to patients still alive after 30 days. Recurrent bleeding was defined as a new episode of bleeding during the period of hospital stay after index bleeding had stopped, manifested as recurrence of symptoms and signs (fresh blood in nasogastric aspirate) of bleeding and this was compared to those without bleeding after index event. The need for surgery was defined as the need to undergo laparotomy after failure of endoscopy interventions to stop bleeding and this was compared to those patients not needing any surgical interventions after index bleed.

### Data and statistical analysis

Data were presented in frequency and percentages unless otherwise stated. Statistical analysis was performed with SPSS version 19.0 (SPSS Inc., Chicago, IL, USA). Univariable and multivariable analyses were used to test the association between variables. Receiver operating characteristics (ROC) curve was utilised to determine the usefulness of Rockall score in predicting the primary outcomes in this study population. A *P* value of < 0.05 was considered statistically significant for all analyses.

## RESULTS

### Incidence of upper GI bleeding and study population characteristics

During the study period between 2003 and 2006, a total of 742 patients (incidence 6.5/100,000 person-years) were registered in the database with a diagnosis of GI bleeding. Of 742 patients, 250 patients (2.2/100,000 person-years) were subsequently identified and confirmed to have non-variceal upper GI bleeding. The incidence of non-variceal bleeding was relatively similar between gender with 1.3/100,000 person-years in men and 1/100,000 person-years in women. Peptic ulcer bleeding was the primary aetiology of non-variceal bleeding in 204 patients (1.8/100,000 person-years or 81.6% of total cases), of which 54% of cases were due to gastric ulcer bleeding (Table 1). Only 2 patients were *H. pylori* positive and both patients were of non-Malays in origin. The mean age of 250

**Table 1** Clinical characteristics of study population.

| Parameters | All | High risk | Mortality | Recurrent bleeding | Need for surgery |
|---|---|---|---|---|---|
| Age, years, mean (SEM) | 62.1 (1.0) | 68.5 (2.6)[#] | 60.5(7.4) | 60.5 (3.4) | 62.4 (4.5) |
| Gender, $n$ (%) | | | | | |
|     Male | 144 (57.6) | 20 (8.0) | 5 (2.0) | 17 (6.8) | 8 (3.2) |
|     Female | 106 (42.4) | 17 (6.8) | 4 (1.6) | 7 (2.8) | 2 (0.8) |
| Ethnic, $n$ (%) | | | | | |
|     Malays | 209 (83.6) | 31 (12.4) | 7 (2.8) | 22 (8.8) | 11 (100) |
|     Non-Malays | 41 (16.4) | 6 (2.4) | 2 (0.8) | 2 (0.8) | 0 |
| Causative lesions, $n$ (%) | | | | | |
|     Peptic Ulcer | 204 (81.6) | 32 (12.8) | 9 (3.6) | 24 (9.6)[#] | 10 (4.0) |
|       Gastric ulcer | 135 (54.0) | 15 (6.0) | 3 (1.2) | 10 (4.0) | 4 (1.6) |
|       Duodenal ulcer | 49 (19.6) | 17 (6.8)[#] | 6 (2.4)[#] | 14 (5.6)[#] | 6 (2.4)[#] |
|       Gastroduodenal ulcers/erosions | 20 (8.0) | 0 | 0 | 0 | 0 |
|     Gastroduodenitis | 36 (14.4) | 4 (1.6) | 0 | 0 | 0 |
|     Others (tumours, telangiectasia) | 10 (4.0) | 1 (0.4) | 0 | 0 | 0 |
| Presenting symptoms, $n$ (%) | | | | | |
|     Melaena | 189 (75.6) | 32 (12.8) | 6 (2.4) | 22 (8.8) | 11 (4.4) |
|     Haematemesis | 117 (46.8) | 16 (6.4) | 4 (1.6) | 9 (3.6) | 3 (1.2) |
|     Epigastric pain | 103 (41.2) | 16 (6.4) | 2 (0.8) | 11 (4.4) | 9(3.6)[#] |
|     Anaemia | 168 (67.2) | 30 (12.0)[#] | 8 (3.2) | 23 (9.2)[#] | 11 (4.4)[#] |
| Laboratory parameters, mean (SEM) | | | | | |
|     Hemoglobin (g/dl) | 8.2 (0.2) | 7.3 (0.4)[#] | 6.5 (0.7)[#] | 6.7 (0.3)[#] | 6.3 (0.4)[#] |
|     Platelet ($\times 10^3$/mm$^3$) | 292.3 (10.4) | 261.9(25.3) | 248.9 (43.1) | 339.7 (50.4) | 375 (51.4) |
|     INR | 1.3 (0.05) | 1.4 (0.1) | 2.0 (0.4) | 1.4 (0.1) | 1.3 (0.1) |
|     aPTT (seconds) | 33.9 (0.8) | 37.9 (1.6)[#] | 39.9 (4.2) | 38.8(3.3) | 35.4 (2.2) |
|     Urea (mmol/l) | 14.1 (0.8) | 18.3 (2.4) | 22.5 (4.5)[#] | 21.0(3.2)[#] | 20.3 (4.4) |
|     Creatinine (mmol/l) | 170.9 (13.6) | 196.3 (35.1) | 316.9 (96.4) | 290.4 (54.4)[#] | 217.8 (54.1) |
| Co-morbidities, $n$ (%) | | | | | |
|     Ischemic heart disease | 53 (21.2) | 14 (5.6)[#] | 2 (0.8) | 1(0.4) | 0 |
|     Chronic renal failure | 41 (16.4) | 12 (4.8)[#] | 2 (0.8) | 7(2.8) | 3 (1.2) |
|     Chronic liver disease | 11 (4.4) | 4 (1.6)[#] | 2 (0.8)[#] | 3 (1.2)[#] | 1 (0.4) |
|     Diabetes Mellitus | 59 (23.6) | 11 (4.4) | 4 (1.6) | 8 (3.2) | 3 (1.2) |
|     Malignancies | 17 (6.8) | 1 (0.4) | 4 (1.6) | 4 (1.6) | 1 (0.4) |
|     Septicaemia | 12 (4.8) | 6 (2.4)[#] | 4 (1.6)[#] | 4(1.6)[#] | 1 (0.4) |
| Risk factors, $n$ (%) | | | | | |
|     Previous peptic ulcer disease | 41 (16.4) | 11 (4.4)[#] | 1(0.4) | 6 (2.4) | 2 (0.8) |
|     NSAIDs | 85 (34.0) | 12 (4.8) | 3 (1.2) | 11 (4.4) | 5 (2.0) |
|     Aspirin | 57 (22.8) | 9 (3.6) | 2 (0.8) | 1 (0.4)[#] | 0 |
|     Clopidogrel | 23 (9.2) | 6 (2.4) | 1 (0.4) | 0 | 0 |
|     Warfarin | 13 (5.2) | 4 (1.6) | 2 (0.8)[#] | 0 | 0 |
|     Corticosteroids | 10 (4.0) | 1 (0.4) | 0 | 2 (0.8) | 1 (0.4) |
|     Herbs/traditional medicine | 4 (1.6) | 2 (0.8) | 0 | 0 | 0 |

[#] significant $P$ value $< 0.05$ (Fisher's exact or Pearson Chi-Square test for categorical and t-test for continuous variables)
$n$, frequency; SEM, standard error of mean.

patients was 62.1 years (range 15–97 years) with older patients, at a mean age of 68.5 years, tended to have a higher risk score ($P = 0.01$).

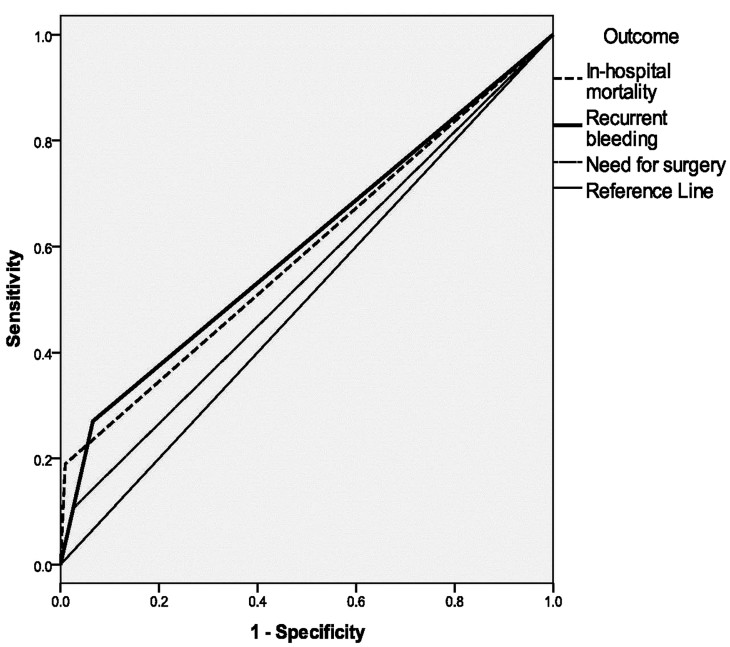

**Figure 1** The usefulness of Rockall score in predicting outcomes in non-variceal upper gastrointestinal bleeding in this ethnic Malay-majority population.

## Primary outcome

The majority of patients were of low risk on admission with 85.2% (213/250) of patients had a Rockall score $\leq 5$ and a mean Rockall score of 4.4. There were 3.6% (9/250) in-hospital mortality, 9.6% (24/250) recurrent bleed and 4.0% (10/250) of patients who subsequently required surgery. A higher Rockall score in this population was associated with increased in-hospital mortality (mean score 7.0, $P < 0.001$), recurrent bleeding (mean score 5.1, $P = 0.01$) and need for surgery (mean score 4.8, $P = 0.01$). A Rockall score $> 5$ was significant in predicting recurrent bleeding in this population but only with area under curve or AUC of 0.6 (95% CI: 0.5–0.7, $P = 0.04$) (Fig. 1).

## Clinical features, co-morbidities and other risk factors

Peptic ulcer bleeding was more likely to re-bleed ($P = 0.04$) during hospitalisation (Table 1). Duodenal ulcers (DU) were more likely to occur in the elderly (mean 66.2 years, $P = 0.04$) but no difference in age was noted with gastric ulcers (GU) (mean 61.1 years with gastric ulcers vs. 63.2 years without gastric ulcers, $P = 0.3$). DUs, but not GUs or gastroduodenal ulcers/erosions, were associated with a higher risk score, mortality, recurrent bleeding and need for surgery (all $P < 0.05$).

Symptoms of anaemia were associated with risk of recurrent bleeding ($P = 0.002$) and need for surgery ($P = 0.02$). Epigastric pain was associated with increased need for surgery ($P = 0.005$). A low hemoglobin level was associated with a higher risk score, in-hospital mortality, recurrent bleeding and need for surgery (all $P < 0.05$). Recurrent

bleeding was more common in those patients with a raised urea ($P = 0.03$) and creatinine ($P = 0.03$). A raised urea was also more likely to be associated with in-hospital mortality ($P = 0.04$).

Both chronic liver disease and septicaemia were significantly associated with increased in-hospital mortality and recurrent bleeding (all $P < 0.05$) (Table 1). History of previous peptic ulcer disease was associated with a higher risk score on admission ($P = 0.02$). More than 1/3 of patients had reported NSAIDs use but on its own, it was not associated with any of the studied outcomes on univariate analysis (Table 1). Aspirin use alone was associated with recurrent bleeding ($P = 0.02$) but warfarin use alone was associated in-hospital mortality ($P = 0.02$) (Table 1).

### Endoscopic features and treatment

Major stigmata of recent hemorrhage (SRH) were present in 26% of all bleeding and their presence were associated with a higher risk score, in-hospital mortality, recurrent bleeding and need for surgery (all $P < 0.05$) (Table 2). More than half were Forrest III lesions (57.2%) and GUs rather than DUs were frequently Forrest III (41.2% vs. 8.4%). However, only Forrest III DUs were associated with recurrent bleeding ($P = 0.04$). DUs were also more likely than GUs to have Forrest I lesions (6.8% vs. 3.6%). Likewise, DUs rather than GUs were associated with a higher risk score, mortality, recurrent bleeding and need for surgery (all $P < 0.05$). GUs were more common than DUs to have Forrest II lesions (9.6% vs. 4.4%) but both were associated with a higher risk score (both $P = 0.01$).

All patients admitted with upper GI bleeding received PPI but there was no difference in studied outcomes between omeprazole and pantoprazole (Table 2). Blood transfusion was needed in 76% of all bleeding and its requirement was associated with risk of recurrent bleeding ($P = 0.001$). Endoscopic interventions were employed in 38.4% of all bleeding, with a third of these were being performed in high risk patients (Table 2). Of all patients with bleeding, adrenaline was the sole intervention in 17.2%, adrenaline with coagulation in 13.2% and adrenaline with clip in 8%. Use of adrenaline only was associated with a higher risk score, recurrent bleeding and need for surgery (all $P < 0.001$). Likewise, adrenaline with clip therapy was associated with a higher risk score, recurrent bleeding and need for surgery (all $P < 0.005$). Adrenaline with coagulation therapy was associated with recurrent bleeding ($P = 0.02$) and need for surgery ($P = 0.005$).

### Multivariable analysis

Of the variables associated with a high Rockall score, major SRH was the factor most predictive of high risk in this population (OR 25.2, 95% CI 8.5–74.3) (Table 3). This variable was also associated with increased in-hospital mortality (OR 11.0, 95% CI 1.9–62.1). Likewise, septicaemia was associated with a high risk score (OR 15.4, 95% CI: 2.9–81.1) and in-hospital mortality (OR 27.1, 95% CI: 4.5-162.8). Warfarin use was the other risk factor associated with in-hospital mortality (OR 16.7, 95% CI 2.1–132.5). Use of adrenaline only during endoscopic intervention was the factor most associated with increased risk of recurrent bleeding (OR 4.4, 95% CI: 1.5–12.7) and need for surgery (OR

**Table 2 Endoscopic features and treatments given.**

| Parameters | All | High risk | Mortality | Recurrent bleeding | Need for surgery |
|---|---|---|---|---|---|
| Stigmata of recent haemorrhage, $n$ (%) | | | | | |
|    None or dark spots | 185 (74.0) | 9 (3.6) | 2 (0.8) | 10 (4.0) | 3(1.2) |
|    Major stigmata | 65 (26.0) | 28 (11.2)[#] | 7 (2.8)[#] | 14 (6.4)[#] | 7 (2.8)[#] |
| Forrest classification, $n$ (%) | | | | | |
|    Forrest I (a: spurting, b: oozing) | 26 (10.4) | 13 (5.2)[#] | 5(2.0)[#] | 11 (4.4)[#] | 6 (2.4)[#] |
|      Gastric ulcer | 9 (3.6) | 3 (1.2) | 1 (0.4) | 3 (1.2) | 3 (1.2) |
|      Duodenal ulcer | 17 (6.8) | 10 (4.0)[#] | 4 (1.6)[#] | 8(3.2)[#] | 3 (1.2)[#] |
|    Forrest II (a: vessel, b: clot, c: haematin) | 35 (14.0) | 13(5.2)[#] | 1 (0.4) | 3 (1.2) | 2 (0.8) |
|      Gastric ulcer | 24 (9.6) | 8 (3.2)[#] | 0 | 2 (0.8) | 1 (0.4) |
|      Duodenal ulcer | 11 (4.4) | 5 (2.0)[#] | 1 (0.4) | 1 (0.4) | 1 (0.4) |
|      Gastroduodenal ulcers | 1 (0.4) | 0 | 0 | 0 | 0 |
|    Forrest III (clean base) | 143 (57.2) | 2 (0.8) | 1 (0.4) | 5 (2.0) | 2(0.8) |
|      Gastric ulcer | 103 (41.2) | 4 (1.6)[#] | 2 (0.8) | 5 (2.0) | 0 |
|      Duodenal ulcer | 21 (8.4) | 2 (0.8) | 1 (0.4) | 5 (2.0)[#] | 2(0.8) |
|      Gastroduodenal ulcers | 19 (7.6) | 0 | 0 | 0 | 0 |
| Type of PPI, $n$ (%) | | | | | |
|    Omeprazole | 42 (16.8) | 6 (2.4) | 0 | 2 (0.8) | 0 |
|    Pantoprazole | 208 (83.2) | 31 (12.4) | 9 (3.6) | 22 (8.8) | 10 (4.0) |
| Tranfusion requirement, $n$ (%) | | | | | |
|    Yes | 190 (76.0) | 32 (12.8) | 9 (3.6) | 24 (9.6)[#] | 10 (4.0) |
|    No | 60 (24.0) | 5 (2.0) | 0 | 0 | 0 |
| Endoscopic intervention, $n$ (%) | | | | | |
|    Adrenaline only | 43 (17.2) | 14 (5.6)[#] | 4 (1.6) | 12(4.8)[#] | 7 (2.8)[#] |
|    + coagulation | 33 (13.2) | 7 (2.8) | 2 (0.8) | 7 (2.8)[#] | 5(2.0)[#] |
|    + clip | 20 (8.0) | 10 (4.0)[#] | 0 | 8 (3.2)[#] | 4(1.6)[#] |

[#] significant $P$ value $< 0.05$ (Fisher's exact test or Pearson Chi-Square test for categorical and t-test for continuous variables)

$n$, frequency.

9.8, 95% CI: 2.3–43.9). Another factor associated with recurrent bleeding was a raised creatinine (OR 1.002, 95% CI: 1.0–1.004). Epigastric pain was highly predictive for increased need of surgery in this population (OR 6.3, 95% CI 1.2–32.2).

## DISCUSSION

In this population starting with an exceptionally low prevalence of *H. pylori* infection, the incidence of 2.2/100,000 person-years of non-variceal upper GI bleeding was also notably low. Peptic ulcer bleeding was the most common cause with an incidence of 1.8/100,000 person-years and this was almost similar to previously reported peptic ulcer perforation of 1.5/100,000 person-years (*Uyub et al., 1994*). The elderly in this population were more susceptible to non-variceal bleeding, especially from DUs and had a higher risk score and concomitant co-morbidities, in keeping with recent observation in *H. pylori*-eradicated populations (*Domon et al., 2012*).

The Rockall score > 5 had been shown to be useful in predicting recurrent bleeding in our population but the AUC result suggests that it may be less satisfactory. This might be

**Table 3 Results of multiple logistic regression analysis (forward: LR).**

| Outcome and risk factors | OR | 95% CI for OR | *P* value |
|---|---|---|---|
| High risk | | | |
|     Major stigmata of bleeding | 25.2 | 8.5–74.3 | < 0.001 |
|     Septicaemia | 15.4 | 2.9–81.1 | 0.001 |
|     Chronic renal failure | 4.1 | 1.3–12.6 | 0.01 |
|     Ischemic heart disease | 3.4 | 1.2–9.7 | 0.02 |
|     Age | 1.05 | 1.0–1.1 | 0.004 |
| In-hospital mortality | | | |
|     Septicaemia | 27.1 | 4.5–162.8 | < 0.001 |
|     Warfarin | 16.7 | 2.1–132.5 | 0.008 |
|     Major stigmata of bleeding | 11.0 | 1.9–62.1 | 0.007 |
| Recurrent bleeding | | | |
|     Adrenaline only | 4.4 | 1.5–12.7 | 0.006 |
|     Creatinine | 1.002 | 1.0–1.004 | 0.04 |
| Need for surgery | | | |
|     Adrenaline only | 9.8 | 2.3–41.9 | 0.002 |
|     Epigastric pain | 6.3 | 1.2–32.2 | 0.03 |

**Notes.**

LR, likelihood ratio; OR, adjusted odd ratio; CI, confidence interval.

related to the threshold of 5 that was initially chosen in the current study. Since many patients admitted to referral institutions were likely to have high risk endoscopic features (stigmata of recent bleed) therefore a threshold of 5 seems reasonable in order to discriminate high risk from the low risk patients, as previously reported (*Bessa et al., 2006*; *Camellini et al., 2004*). However, our results reported a lower rate of high risk features, which would then have resulted in a less accurate AUC. Hence, a threshold below 5 may have been preferable in our population and this merits further analysis. A number of recent studies have similarly questioned the accuracy of Rockall scores in different populations and settings (*Wang et al., 2013*; *Custódio Lima et al., 2013*). Further prospective studies are needed to verify their findings.

Our study shared similar baseline characteristics with *Vreeburg et al. (1999)* including definition for mortality but our results suggest a better prediction of recurrent bleeding rather than in-hospital death. A higher rate of recurrent bleeding observed in our population as compared to in-hospital mortality might explain this discrepancy. The low hemoglobin and urea levels indicate a minor bleeding risk in general, compatible with the overall low risk score observed in this population. The generally low risk score in this population would not, however, allow one to decide for the need of therapeutic endoscopy. The Blatchford score may have been more useful in this regard (*Pang et al., 2010*).

Among the variables described in Rockall score, SRH stood out as the most predictive of high risk and in-hospital mortality. The presence of SRH is of greater significance in *H. pylori*-associated bleeding GUs than DUs (*Chang-Chien et al., 1988*). In our study population, more than half of upper GI bleeding was a result of GUs with only 20% due to DUs. However, GUs were more likely Forrest III lesions (57.2%) but had relatively benign

outcomes. In contrast, DUs, while less common, and were more likely Forrest I and II lesions, but there was significant associations between DUs with all studied outcomes. Previous studies have also similarly observed that *H. pylori*-negative DUs were more likely to bleed and were more common among the elderly population with risk factors (*Gisbert & Calvet, 2009*; *Chu et al., 2005*). These studies were limited by false negative results for *H. pylori*, but our study population did not suffer from this limitation (*Gisbert & Calvet, 2009*).

Septicaemia, while not a variable in the Rockall score, was also highly predictive of high risk and in-hospital mortality, similarly reported by Zimmerman and others (*Zimmerman et al., 1995*; *Afessa, 1999*). In the original Rockall validation study, pneumonia which was associated with septicaemia was included in the model but not in the complete model (*Rockall et al., 1996*). Our study suggests that septicaemia, if present, should be considered as a major co-morbidity and should be given a score of 2. An elevated creatinine was predictive of increased risk of recurrent bleeding in the multivariable analysis, in agreement with Zuckerman and others (*Zuckermann et al., 1985*; *Sood et al., 2012*). Patients with ischemic heart disease, the most common co-morbidity in our study, were also frequently impaired in their renal function (*Shalev et al., 2012*).

Non-variceal upper GI bleeding was associated with more adverse outcomes in the current study with a mostly elderly population, and an almost absence of *H. pylori* infection, and in the presence of offending agents including aspirin, NSAIDs and warfarin. In a study from Japan, the usage of aspirin and NSAIDs was not associated with a serious outcome in GU bleeding (*Ishikawa et al., 2008*) but the role of *H. pylori* infection was not addressed. Recent studies found that patients with *H. pylori*-negative peptic ulcers and who took aspirin were more likely to have a higher bleeding risk (*Kang et al., 2011*; *Chan et al., 2013*; *Hernandez-Diaz & Garcia Rodriquez, 2006*). NSAIDs and aspirin in combination, rather in isolation, would have produced the greatest risk, and this might explain why our multivariate analysis did not have any association (*Hreinsson et al., 2013*; *Lee & Noridah, 2013*). Further studies are needed to determine the significance of our findings since these results may have a long term impact in the *H. pylori*-eradicated populations.

Endoscopic interventions were carried out in only a third of patients with high risk scores and this implied that the Rockall score is not useful in selecting patients who require interventions. Endoscopic therapy with adrenaline only was associated with a four-fold risk for recurrent bleeding and a ten-fold risk for surgical intervention based on the multivariable analysis; figures that were similar to *Levin et al. (2012)*. This indicates that adrenaline alone is unlikely to be sufficient when endoscopic interventions are needed and is not in line with our current practice or guidelines (*Chung et al., 1997*; *Chung et al., 1999*).

The need for surgery was not an outcome initially included during the validation study of the Rockall score, however surgical intervention is frequently sought in the setting of failed endoscopic therapies. In the current study, the need for surgical intervention of 4.0% was relatively similar to the rate of in-hospital mortality of 3.6%. Previous study

indicated an overall mortality of 34.1% in patients with upper GI bleeding requiring surgery (*Czymek et al., 2012*). Epigastric pain, predictive of the need of surgery, may be a sign of impending perforation, and should be carefully assessed especially in this largely rural-majority population who often present late in their course of disease.

Study limitations should be mentioned. There was only one other centre in the region managing upper GI bleeding, and this might affect our incidence calculation slightly but our institution is a primary referral institution with a reliable GI database. The rate of *H. pylori* infection was underestimated in this study for several reasons. One was the use of rapid urease test, which had been shown to be unreliable during the acute phase of ulcer bleeding (*Schilling et al., 2003*; *Lee et al., 2000*). Due to this reason, based on the discretion of the managing endoscopist, only 35.6% of patients were tested. Of those tested negative with the urease test, histology was reliable in confirming the absence of *H. pylori* without other additional tests. On the other hand, this limitation on *H. pylori* would not have affected our current study since we have already published extensively on the exceptional low prevalence of *H. pylori* in the region (*Lee, Mahendra Raj & Graham, 2013*; *Uyub et al., 1994*; *Raj et al., 2001*).

So what would be the future in patients having non-variceal upper GI bleeding in the absence of *H. pylori*? We can conclude that based on our population with an exceptional low prevalence of *H. pylori* infection and also peptic ulcer disease, acute non-variceal upper GI bleeding is also of low incidence, similar to peptic ulcer perforation rates. An absence of *H. pylori* infection may not however reduce the risk of peptic ulcer bleeding in the presence of risk factors especially offending drugs in an elderly population.

## ACKNOWLEDGEMENTS

We are grateful to Professor David Graham from the Baylor College of Medicine, Houston, Texas for his valuable comments to the paper.

### Funding

We would like to thank Universiti Sains Malaysia for the incentive grant in 2007. The funders had no role in study design, data collection and analysis, decision to publish, or preparation of the manuscript.

### Grant Disclosures

The following grant information was disclosed by the authors:
Universiti Sains Malaysia; incentive grant in 2007.

### Competing Interests

The authors declare that they have no competing interests.

### Author Contributions

- Yeong Yeh Lee and Nordin Noridah conceived and designed the experiments, performed the experiments, analyzed the data, contributed reagents/materials/analysis tools, wrote the paper.

- Syed Abdul Aziz Syed Hassan and Jayaram Menon conceived and designed the experiments, wrote the paper.

## Supplemental Information

Supplemental information for this article can be found online at http://dx.doi.org/10.7717/peerj.257.

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
