# Peer review of "Absence of Helicobacter pylori is not protective against peptic ulcer bleeding in elderly on offending agents: lessons from an exceptionally low prevalence population"

_PeerJ, doi:10.7717/peerj.257_

## Round 0.1 · original submission · Minor Revisions

· Academic Editor

Minor Revisions

Both reviewers found your manuscript was sound and has scientific merit, and raised a number of questions which need to be addressed.

Reviewer 1 ·

Basic reporting

The basic reporting style is adequate. However, the english grammar is not. Further correction of grammar is necessary before this can be considered acceptable for publication.

Experimental design

This is a prospective audit of GI bleeding from a single centre in a region of the world with a low prevalence of Helicobacter pylori infection. The study design is satisfactory.

Validity of the findings

1. The incidence of peptic ulcer bleeding is based on the assumption that all/ most GI bleeding in this region is managed at this single centre. This is not the case & the authors should clarify this with some relevant statistics :
i) how many other medical centres are likely to be managing GI bleeding in this region ?
ii) what proportion of GI bleeding is managed at this single centre ?
iii) a limitation to this “incidence calculation” should be mentioned in the discussion

2. The diagnosis of H. pylori infection during an acute GI bleed using the standard “CLO” test has been shown to be unreliable. Hence, there may be a possibility of a false negative rate for the H. pylori diagnosis. Please clarify:
i) What proportion of patients had a “CLO” test ?
ii) Were additional tests, eg urea breath tests, performed on those tested negative for CLO ?
Ref:

a) A negative rapid urease test is unreliable for exclusion of Helicobacter pylori infection during acute phase of ulcer bleeding. A prospective case control study. Schilling D, Demel A, Adamek HE, Nüsse T, Weidmann E, Riemann JF.
Dig Liver Dis. 2003 Apr;35(4):217-21.

b) Rapid urease tests lack sensitivity in Helicobacter pylori diagnosis when peptic ulcer disease presents with bleeding. Lee JM, Breslin NP, Fallon C, O'Morain CA.
Am J Gastroenterol. 2000 May;95(5):1166-70.


3. The “secondary outcomes” on this study is mostly a description of the clinical features and outcomes of the study subjects. This is not really an “outcome measure” and should be changed accordingly

4. The authors have demonstrated that the Rockall score had a low accuracy (AUC 0.6) for predicting clinically important outcomes such as re-bleeding, etc. However, this is not well explored in the ‘Discussion” section. I would suggest a paragraph on possible explanations for the low accuracy of the Rockall score & whether this observation has been reported elsewhere.

5. NSAIDs/ Aspirin was not found to be associated with any of the clinically relevant outcomes measured in the multi-variate analysis – is there an explanation for this ?

Additional comments

1. The general English grammar of the manuscript needs to be improved. I would suggest using Microsoft Word to help with this.

2. Single modal therapy (i.e. Adrenaline only) during endoscopic intervention was found to be associated with important outcomes. This is not surprising, as there is already well established evidence & practice guidelines that Adrenaline alone is inferior to dual therapy for bleeding peptic ulcer disease.
I would suggest modifying your “Discussion – lines 219 – 224” to reflect this & that the use of Adrenaline only is not in line with current practice.

3. Table 1 & 2 should be combined.

Reviewer 2 ·

Basic reporting

This is an excellent manuscript from a group that has a long history of studying effect of Helicobacter pylori in gastroenterology. The manuscript is sound in terms of its analysis and methods, extending the understanding of the risk factors of peptic ulcer bleeding. Specifically, H. pylori infection may not be the risk factor of peptic ulcer bleeding in the presence of risk factors especially offending drugs in the elderly.

Experimental design

The methods are sound. I only ask the authors to expand a bit and explain their choice for hospital (Hospital XXX, page 4, line 65).

Validity of the findings

The conclusions were generally made from solid observation and analysis.

Additional comments

I noted few errors in the text, please proof-read again carefully.

---

## Round 0.2 · accepted · Accept

· Academic Editor

Accept

Thanks for submitting good papers to PeerJ. The references need to be reformated according to PeerJ style.